# Modulating the Tumour Microenvironment by Intratumoural Injection of Pattern Recognition Receptor Agonists

**DOI:** 10.3390/cancers12123824

**Published:** 2020-12-18

**Authors:** Olivia K. Burn, Kef K. Prasit, Ian F. Hermans

**Affiliations:** 1Malaghan Institute of Medical Research, P.O. Box 7060, Wellington 6042, New Zealand; oburn@malaghan.org.nz (O.K.B.); kprasit@malaghan.org.nz (K.K.P.); 2Maurice Wilkins Centre, Private Bag 92019, Auckland 1042, New Zealand

**Keywords:** pattern-recognition receptors, toll-like receptors, intratumoural, tumour microenvironment

## Abstract

**Simple Summary:**

The immune system is capable of eliminating solid cancers through the action of immune cells that recognise antigens that are unique to tumour tissue. However, the activity of tumour-specific immune cells is often blunted by the immunosuppressive environment within the tumour core. One strategy to overcome this limitation is to inject immune modulators directly into the tumour bed to stimulate the local network of immune cells. Not only does this promote local antitumour activity, but also facilitates the infiltration of immune cells with antitumour activity at distant tumour sites. A major class of compounds used for this purpose are recognised by pattern recognition receptors (PRR), providing molecular cues typically associated with infection or tissue damage to inflate the response. In this review, we summarise research into the use of such compounds in preclinical studies, including promising studies conducted in combination with conventional cancer therapies and other immunotherapies.

**Abstract:**

Signalling through pattern recognition receptors (PRRs) leads to strong proinflammatory responses, enhancing the activity of antigen presenting cells and shaping adaptive immune responses against tumour associated antigens. Unfortunately, toxicities associated with systemic administration of these agonists have limited their clinical use to date. Direct injection of PRR agonists into the tumour can enhance immune responses by directly modulating the cells present in the tumour microenvironment. This can improve local antitumour activity, but importantly, also facilitates systemic responses that limit tumour growth at distant sites. As such, this form of therapy could be used clinically where metastatic tumour lesions are accessible, or as neoadjuvant therapy. In this review, we summarise current preclinical data on intratumoural administration of PRR agonists, including new strategies to optimise delivery and impact, and combination studies with current and promising new cancer therapies.

## 1. Introduction

There have been significant advances in the development and use of immunomodulatory compounds to reverse immune tolerance and exhaustion in order to facilitate anti-tumour immune responses. While the most clinically relevant advance has been the development of immune checkpoint inhibitors to block negative signalling in anti-tumour T cells, other approaches have focused on delivering stimulatory signals, either to T cells directly, or to the antigen-presenting cells (APCs) that ultimately orchestrate adaptive immunity. One broad class of such immunostimulatory compounds are agonists for patter recognition receptors (PRRs), a series of innate receptors that screen the environment for molecular cues associated with infection or tissue distress [1]. Integration of signals facilitated by these innate immune-sensing pathways plays a significant role in regulating immediate host innate immunity, and also shaping the quality of induced adaptive responses featuring T cells and B cells [2]. 

It is now recognised that a critical barrier to effective immunity against solid tumours is the local tumour microenvironment (TME), with its unique local milieu of immunosuppressive cells and hypoxia-induced factors that ultimately serve to limit the activity of infiltrating immune effector cells [3,4,5]. By administering PRR agonists to the host, it is possible to deliver positive signals that can overcome some of these immunosuppressive networks [6,7,8]. Furthermore, the administration of PRR agonists into an environment where APCs have captured tumour antigens can create an immunopermissive environment where antigen-specific adaptive responses can be initiated, or existing ones given an effective boost [9]. To maximise the likelihood of stimulating appropriate antigen-loaded APCs, it may be necessary to introduce the PRR agonists directly into the TME. This can be achieved in some clinical situations by intratumoural administration, the focus of this review.

Many PRRs have been described, with the main families explored in the context of intratumoural administration being the Toll-like receptors (TLR), retinoic acid-inducible gene I (RIG-I)-like receptors (RLRs), and DNA sensors such as stimulator of interferon genes (STING). All of these receptors transmit signals when they encounter agonists in the form of pathogen-associated molecular patterns (PAMPs) or damage-associated molecular patterns (DAMPs) [10,11]. PAMPs are structures typically observed in pathogenic prokaryotes, such as lipopolysaccharide (LPS), peptidoglycans, and lipoproteins derived from bacterial cell walls, bacterial structural proteins such as flagellin, nucleic structures such the unmethylated deoxyribonucleic acid (DNA) associated with bacteria and some fungi and parasites, or double stranded ribonucleic acid (dsRNA) typically seen in viral species [12]. DAMPs result from cellular stress, apoptosis, and necrosis, such as chromatin-associated protein high-mobility group box 1 (HMGB1) [13], heat shock proteins (HSPs) [14], extracellular adenosine triphosphate (ATP) [15], and proteolytically digested fragments of the extracellular matrix, such as biglycan and fibrinogen [16,17]. Although their intracellular signalling pathways are often distinct, they ultimately lead (alone or in combination) to increased transcription of genes involved in inflammatory responses. These include proinflammatory cytokines such as IL-1, IL-6, IL-12, and tumour necrosis factor (TNF), type-I interferons (IFNs) that promote APC maturation and the cytotoxic activity of macrophages and natural killer (NK) cells [18,19,20,21,22], and chemokines that recruit lymphocytes and neutrophils [23,24]. Combined, these factors contribute to the mobilisation of adaptive immune responses. While PRR agonists have been investigated as systemic therapies, advancement to the clinic has been hampered by systemic toxicities [25,26,27]. However, studies suggest that an intratumoural route of administration can lead to high local concentrations of PRR agonists in the TME while limiting systemic exposure that can potentially lead to toxicity [6,28], even when used in a repetitive dosing regimen [29,30,31].

While PRRs are highly expressed by APC lineages that are known to infiltrate the TME, including conventional myeloid dendritic cells (cDC), plasmacytoid dendritic cells (pDC), and tumour associated macrophages (TAMs), they can also be expressed on CD8^+^ and CD4^+^ T cells, regulatory T cells (Treg), and B cells [32]. The impact of PRR agonism on the adaptive immune response can therefore be indirect, through the activation of APCs, or direct, through direct stimulation of adaptive effectors cells. Importantly, although the immediate response to intratumoural delivery of an agonist is to alter the function of cells in the local TME, numerous studies have shown that it is possible to create an environment that supports a systemic immune response that can initiate tumour regression in non-injected distal tumour sites, referred to as an abscopal effect. Here we discuss the recent advances in our understanding of how the TME can be shaped by intratumoural PRR agonists (Figure 1). In much of this preclinical work, PRR agonists have been studied in combination with other immunomodulatory strategies, such as with checkpoint inhibitors, or with conventional therapies such as radiation and chemotherapy, in order to promote long-term tumour eradication (summarised in Table 1).

## 2. Intratumoural PRR Ligands and the TME

### 2.1. TLR Agonists

Downstream TLR signalling involves the recruitment of adapter proteins, with myeloid differentiation factor 88 (MyD88) utilised by all TLRs excluding TLR3, and TIR domain-containing adapter inducing interferon β (TRIF) used by both TLR3 and TLR4. Engagement of these adapters ultimately transmits signals through nuclear factor-κB (NF-κB), mitogen-activated protein (MAP) kinases, and interferon-regulatory factors (IRFs) to influence the expression of genes relevant to inflammation (reviewed extensively elsewhere; [54]). When considering APCs such as DCs, signalling through TLRs expressed on the cell surface (TLR1, 2, 4, 5, 6) generally induces a cytokine profile featuring IL-1, IL-12, and TNF, while TLRs within the endosomal compartment (TLR3, 7, 8, 9) promote significant type I IFN release [55,56,57]. Through direct signalling, or via the cytokines induced, TLRs can regulate essential processes for initiating T cell immunity, including altering rates of antigen uptake, processing and presentation of antigen by APCs, as well as inducing enhanced expression of molecules required for T cell activation, including the critical co-stimulatory molecules of the B7 family (e.g., CD80 and CD86) and TNFR family (e.g., CD40, OX40L) [58,59,60,61,62]. Below, we review the effect of agonists for the different TLRs in the context of intratumoural cancer treatment. Although TLR signalling occurs in cells that are not regarded as APCs, much of the data indicates a critical role for enhancing APC function in the efficacy of these treatments. 

#### 2.1.1. TLR9

Perhaps the most clinically advanced of the PRR agonists evaluated target TLR9 (reviewed in [63,64]). In general, monotherapy with intratumoural administration has shown to be safe, but with limited clinical efficacy observed to date, although more promising results have been observed when injected in combination with other cancer therapies. Further preclinical studies, like those described below, are needed to optimise this therapy for the clinic.

Primarily expressed on B cells and pDCs, TLR9 is located in intracellular vesicles, where it detects the presence of unmethylated cytidine phosphate guanosine (CpG) dinucleotides associated with the genomes of most bacteria and DNA viruses, inducing signalling via the MyD88 pathway [65]. Synthetic TLR9 agonists have been developed based on CpG-oligodinucleotides (ODN) containing hexamer CpG motifs that are not common in vertebrate DNA. Early studies with intratumoural CpG-ODNs in Lewis rats implanted with syngeneic glioma cells showed increased tumour infiltration with macrophage/microglial cells, CD8^+^ T cells, and NK cells, and protection against a second tumour challenge. In follow-up studies in mice, the effect was lost in nude mice, suggesting that CpG-ODN was not directly cytotoxic and required immunostimulation for the antitumour effect [66]. Specific cell depletion studies showed that macrophages played a critical role in the early phase of tumour rejection, capable of controlling growth temporarily in nude mice, but full rejection required a later phase only seen in T cell-replete animals [67]. Several studies in different tumour models in mice confirmed a critical role for CD8^+^ T cells, and establishment of a memory response [68], with transfer of activated T cells sufficient to mount fully protective responses in RAG-1 knockout mice (which are otherwise deficient in adaptive immune cells) [69]. Initial priming was shown to involve modulation of DC function in a CT26 colon carcinoma model, with CpG-ODN treatment shown to induce the expression of CCL20 in the tumour, attracting large numbers of circulating DCs into the tumour mass. Whereas the TME typically caused inhibition of DC activation in this model, CPG-ODN treatment enabled tumoural DCs to effectively cross-present tumour antigens to activate CD8^+^ T cells. This local effect on tumoural DC number and function was critical, as systemic delivery of the DC growth factor, Flt3 ligand could increase circulating DCs, but had no effect on the number of tumoural DCs or impact on tumour growth [70]. More mechanistic insight came from studies comparing intratumoural CpG-ODN to intravenous treatment, where intratumoural treatment was shown to lead to more extensive infiltration of antigen-specific T cells, which was attributed to significantly higher levels of inflammatory chemokines (RANTES, IP-10, MCP-1, MCP5, MIP1α, and MIP1β) in the TME. In vivo, depletion of pDCs, which are known to express TLR9 [71,72], greatly reduced the levels of chemokines induced, and impaired T cell accumulation and the antitumour effect [73]. Given the known capacity for pDCs to produce type I IFNs, this cytokine response was likely key to the improved T cell activity, through its effects on APC function, including improved cross-priming [74,75,76], and also its direct effect on T cell memory formation [77,78]. Indeed, a type I IFN response is now a feature used in developing this intratumoural therapy. For example, in a recent study, twenty novel CpG-C ODNs were screened for their ability to induce secretion of IFN-α (along with IL-6 and TNF) in human peripheral blood mononuclear cells (PBMCs) to select new efficacious agonists for further evaluation [79]. Interestingly, one study did not show a role for pDC in response to CpG-ODN in a mouse model of T cell lymphoma; instead, CpG-B ODN stimulation recruited neutrophils into the milieu, resulting in the activation of cDCs, with subsequent increased antitumour T cell priming in draining lymph nodes [80]. Another important facet of the intratumoural effect of CpG-ODN is the impact on myeloid-derived suppressor cells (MDSC), which represents an important constituent of the immunosuppressive TME. Delivery of CpG-ODN directly into the tumour has been shown to reduce the immunosuppressive activity of monocytic MDSC (which express TLR9) and cause their differentiation into macrophages with direct tumouricidal capability [81].

Intratumoural CpG-ODN treatment has been evaluated in combination with evolving new therapies based on antibody-mediated modulation of T cells. Intravenously administered monoclonal antibodies against OX40, which enhance T cell activation, or against CTLA4 to block a cell-intrinsic negative regulatory immune checkpoint in T cells, combined effectively with the intratumoural CpG-ODN treatment in a mouse model of B cell lymphoma [40]. In fact, B cell lymphoma may represent an unusual case, as it was unexpectedly found that tumour rejection did not require host expression of TLR9, and was associated with increased expression of the co-stimulatory molecules CD80 and CD86 on the lymphoma cells; the lymphoma cells themselves may therefore be acting as APCs [82]. In some studies, CpG-ODN induced the upregulation of PD-1 expression on immune cells [83], making this an obvious, clinically relevant combination to explore. Durable rejection of injected tumours, and activity against un-injected distant tumours, was observed when CpG-ODN was combined with anti-PD-1 in models of CT26 colon carcinoma, TSA mammary adenocarcinoma, and MCA38 colon carcinoma—all models that showed little response to PD-1 blockade alone [44]. The effect of PD-1 blockade was to alter CpG-ODN-mediated differentiation of tumour-specific CD8^+^ T cells into short-lived effector cells, and preferentially expand long-lived memory precursors, which likely accounted for the increased durability of the combination [44]. Studies of CpG-ODN treatment with anti-CTLA-4 in B16 melanoma were also encouraging, although when evaluated further, only the more immunogenic model, B16.OVA, resulted in full synergy with abscopal effect, whereas in the less immunogenic B16.F10 melanoma model, only the locally injected site responded to the combination [41]. Combining with a CTLA-4 antibody of increased potency improved the distant response. Interestingly, in some studies, CpG-ODN administration was found to increase OX40 co-stimulatory receptor expression on CD4^+^ T cells, and combining intratumoural treatment with intravenous agonistic anti-OX40 antibody was found to be particularly effective [42]. Successful combination therapy with anti-OX40 and anti-CTLA-4 could be achieved by administering the antibodies intratumourally, which helped deplete immunosuppressive Tregs in the TME [43]. The upregulation of OX40 has been investigated as an early predictor of response to CpG-ODN, with increased uptake of a near-infrared (NIR) fluorescence probe affixed to an antibody for OX40 seen with CpG-ODN treatment in a model of hepatocellular carcinoma [84]. In the clinic, checkpoint blockade is currently being evaluated in combination with other conventional therapies, such as radiation treatment [85]. However, promising preclinical studies indicate that intratumoural CpG-ODN treatment could be added to the mix. For example, local radiotherapy, intratumoural CpG-ODN, and systemic PD-1 blockade were shown to be particularly effective in lung cancer models, associated with increased DC activation and infiltration of cytokine-producing CD8^+^ T cells [45]. 

Low-dose metronomic cyclophosphamide complements the actions of an intratumoural CpG-ODN to potentiate innate immunity and drive anti-tumour responses [31]. In this setting, the chemotherapy shaped the immune response, with a reduction in Tregs, and an increase in M1-type macrophages, a phenotype typically associated with antitumour effects through direct tumouricidal activity and release of factors such as IL-12, CXCL9, and CXCL10 that enhance T cell and NK cell function and migration to the tumour. A corresponding decrease in M2-type macrophages was observed. These pro-tumourigenic cells produce the immunosuppressive cytokine IL-10, are able to recruit Tregs, and express PD-L1, which triggers negative signalling via PD-1 in T cells; they also promote angiogenesis and tumour progression through the production of VEGF and the release of chemokines such as CCL18 and CCL22, and enzymes like matrix metalloproteinase [86,87,88]. Another way to exploit chemotherapeutic agents in combination therapy is to use agents such as doxorubicin and oxaliplatin that induce immunogenic cell death (ICD) to release antigens and broaden the immune response. A recent study used intratumoural injection of an ICD inducer in combination with a cell-labelling version of CpG-ODN that is retained at the injection site to generate an in situ apoptotic cell-adjuvant complex to drive antitumour responses [46]. Ibrutinib, an irreversible inhibitor of Bruton’s tyrosine kinase, that is an effective treatment against many types of B cell lymphomas, also causes ICD and was shown to combine effectively with intratumoural CpG-ODN in a mouse model of lymphoma [47]. 

Many recent preclinical studies have focused on improving the delivery of CpG-ODN to the TME and have often been conducted in combination with other therapies. Potent anti-tumour responses were observed in mice by incorporating CpG-ODN and anti-CD40 agonist antibody into liposomal carriers, with elimination of systemic toxic side effects achieved by focused sequestration in the TME [89]. As the phosphodiester-backbones of CpG-ODN can be susceptible to nuclease activity, encapsulation into liposomes has been used to improve stability in vivo [90]. This was also achieved using a poly(L-glutamic acid)-modified CpG-ODN [91], or modified with polyethyleneimine [92]. Conjugating CpG-ODN to carbon nanotubes proved to be effective in intracranial treatment, abrogating tumour growth not only in the brain, but also subcutaneous tumours [93]. In a recent study, the challenge of delivering a payload into the compressed TME was overcome by loading CpG onto apoptotic bodies, which were phagocytosed by inflammatory monocytes that actively infiltrate the tumour centre [94]. Others have utilised delivery on synthetic high density lipoproteins (together with a chemotherapeutic agent) [95] or used the specific targeting of some peptide moieties for the TME to direct uptake into the tumour [96]. An albumin-binding analogue of CpG-ODN was shown to accumulate in tumours after the local tumour area was subjected to radiation therapy [97]. In some studies, the delivery vehicles were functionalised to provide additional antitumour activity. For example, CpG-ODN and tumour antigens were integrated with gold nanorods capable of photo-heat conversion, so that illumination of the tumour area with NIR light resulted in a local rise in temperature that caused additional tumour destruction [98]. CpG-ODN has also been incorporated into carbon nanotubes, which have a photohyperthermic effect induced by NIR irradiation [99], and a similar concept has been used with a hydrogel containing gold nanoparticles exposed to laser irradiation at 780 nm to cause the local temperature rise [100]. Tumour elimination with a low radioactivity dose was achieved by intratumoural injection of a sodium alginate formulation containing CpG-ODN and a catalase labelled with the therapeutic ^131^I radioisotope [101].

The CpG-ODNs evaluated preclinically fall into three general classes. Class-A CpG ODNs stimulate pDC to produce high quantities of IFN-α, induce maturation of APC, and activate NK cells indirectly via the IFN-α produced. Class-B CpG-ODNs activate B cells and NK cells, but are associated with lower levels of DC activation and IFN-α induction, while Class-C CpG-ODNs combine the properties of class-A and Class-B. Toxicities associated with repeated administration (with systemic access) can cause splenomegaly, lymphoid follicle destruction, and hepatotoxicity in mice [102,103,104]. Some hematologic adverse events and influenza-like symptoms have been observed in human cancer clinical trials [105,106], and some adverse events, such as platelet activation, complement activation, and clotting time prolongation are caused by the unnatural phosphorothioate backbone used in Class-B CpG-ODNs [107,108,109]. In contrast, Class-A CpG-ODNs are composed of mostly natural phosphodiester backbones, but as such are susceptible to degradation [110]. To address this, in a recent study Class-A CpG-ODN was encapsulated into lipid nanoparticles, which showed potential as a safe and effective alternative formulation for cancer immunotherapy [111]. Another downside of Class-B CpG-ODNs is that they can limit their immune stimulatory activity through activation of the signal transducer and activator of transcription 3 (STAT3), which is responsible for orchestrating immunosuppressive networks in the TME. A recent study showed in vitro that while Class-B CpG-ODN alone can induce activation of DCs, combined inhibition of the JAK2/STAT3 pathway resulted in superior DC activity [112].

Another strategy to improve intratumoural responses to intratumoural administration involves combining TLR signalling via both the TRIF and MyD88 pathways to activate separate kinetics. It has been shown that combining CpG-ODN with the TLR3 agonist poly I:C can synergistically upregulate the expression of IL-12p40 in murine peritoneal macrophages, and potentially lead to enhanced antigen cross-presentation [113]. Intratumoural administration of both poly I:C and CpG-ODN, combined with systemic transfer of melanoma-specific T cells, led to the eradication of established tumours [35], which was attributed to improved activity of host DCs. The improved T cell activity was lost in Tap-deficient mice, suggesting an enhanced capacity for cross-presentation was essential to the response. In another study involving co-delivery of agonists, concurrent targeting of TLRs 7, 8, and 9 by intratumoural injection increased the number and tumouricidal activity of tumour infiltrating CD8^+^ T cells and NK cells, and reduced the frequency of immunosuppressive MDSCs [48].

#### 2.1.2. TLR2

TLR2 recognises bacterial-derived lipopeptides and signals as a heterodimer with either TLR1 or TLR6. The activity of Bacillus Calmette-Guerin (BCG), a clinically approved intratumoural therapy for bladder cancer, has been attributed to a number of PAMPs, including the activation of TLR1/TLR2 signalling via recognition of lipopeptides within the cell wall [114]. Pam_3_CysSerLys_4_ (Pam_3_CSK_4_) is a triacylated lipopeptide that mimics the acylated amino terminus of bacterial lipoproteins and is commonly used as a synthetic TLR1/TLR2 agonist [115]. Intratumoural administration of Pam_3_CSK_4_ alone has been attempted, but had no significant effect on tumour growth in a B16.F10 melanoma model [33]. A recently described TLR2-specific agonist based on a glucomannan polysaccharide modified with acetyl groups (acGM-1.8) displayed a much higher safety profile than Pam_3_CSK_4_, with some antitumour activity as a single agent [116]. Following intratumoural administration of acGM-1.8 into mice with established sarcoma lesions on both flanks, strong anti-tumour responses were observed against both the treated and untreated bilateral tumours. The local response was abrogated by intratumoural clodronate liposome injection, highlighting a role for macrophage activity. In support of this, large changes in the phenotype of TAMs were observed, with an increase in M1-type macrophages and decrease in M2-type macrophages. In fact, intratumoural acGM-1.8 changed the overall cytokine profile within the tumour, with an increase in levels of TNF, IL-12p70, and IFN-γ, and an increased ratio of effector T cells (Teff) to Tregs. Furthermore, the antitumour response was not induced in Rag^−/−^ mice, suggesting induction of a crucial adaptive immune response [116].

Studies in mice conducted with intratumoural Pam_3_CSK_4_ combined with systemic administration of anti-CTLA-4 showed enhanced efficacy in a B16.F10 melanoma model [33]. Interestingly, in this combination setting, there was an increase in Fcγ receptor IV expression on macrophages, resulting in greater antibody-dependent macrophage-mediated depletion of CTLA-4-positive Tregs in the tumour [33,117]. Thus, intratumoural delivery of PRR agonists can play unexpected supporting roles in established treatments, and carefully selected combinations can result in synergistic effects. 

#### 2.1.3. TLR3

TLR3 recognises exogenous and endogenous dsRNA in endosomes, stimulating a signalling pathway via TRIF that culminates with the secretion of inflammatory cytokines and type I IFN. Poly I:C is an analogue of viral dsRNA that acts as an agonist of endosomal TLR3 [118]. In mice, intratumoural administration of poly I:C delayed B16.OVA melanoma and E0771 breast cancer outgrowth [83]. Moreover, intratumoural administration of a nanoparticle form of poly I:C, BO-112, induced strong anti-tumour activity in multiple mouse tumour models where subcutaneous administration failed [119]. The antitumour effects were associated with enhanced infiltration of antigen-specific T cells into the tumour and tumour-draining lymph nodes, with an increased Teff/Treg ratio. The protection provided by either intratumoural BO-112 or poly I:C was lost in Batf3^−/−^ mice, which lack conventional DC-1 cells (cDC1), a population with a heightened propensity for cross-priming CD8^+^ T cells, that was shown to be the major TLR3-positive DC population in the tumour tissue in wild-type animals [34,119,120]. The anti-tumour effects of the TLR3 agonists were also abolished in IFNAR1 knockout mice [34,119,120], highlighting a key role for type I IFN signalling.

Intratumoural administration of poly A:U, a double-stranded polyribonucleotide that is less toxic (but also less potent) than poly I:C, could also delay the growth of B16 melanomas, which was again associated with an increased Teff/Treg ratio, and also a greater number of CD8^+^ T cells expressing the effector molecule granzyme B [120,121,122]. There was also increased expression of PD-1 on CD8^+^ T cells and PD-L1 on myeloid cell populations including M2-type macrophages, suggesting combinations of poly A:U with inhibitors to neutralise this immune checkpoint may enhance the response [120]. Combining intratumoural poly I:C with systemic administration of Flt3 ligand resulted in enhanced cDC1 accumulation in the tumour and was associated with an increased tumour antigen-specific CD8^+^ T cell response. Taking this a step further, responses were improved again when Flt3 and poly I:C were used in conjunction with PD-L1 blockade and B-Raf proto-oncogene (BRAF) blockade [34], two commonly used treatments in the clinic. Overall, given that combination treatments have been more effective in preclinical studies, intratumoural TLR3 agonists like BO-112 are currently being investigated in patients with solid tumours in combination with systemic anti-PD-1 (NCT02828098; NCT04508140; NCT02423863) [123].

#### 2.1.4. TLR4

Agonists for TLR4 are already clinically approved for use as immune adjuvants in vaccine therapy, as exemplified by the use of monophosphoryl lipid A (MPL), a detoxified version of the well-known TLR4 agonist LPS, in the human papillomavirus (HPV) vaccine, Cervarix^TM^ [124,125]. Preclinical studies in mice have shown that intratumoural administration of LPS can result in antitumour effects, with more consistent tumour control observed compared to subcutaneous administration at a contralateral site or intraperitoneal administration [36,126,127]. Assessment of tumours after intratumoural treatment with LPS, or with a synthetic lipid A derivative (formulated glucopyranosyl lipid A; G100), revealed an inflamed tumour milieu, with an influx of effector T cells and NK cells and increased expression of activation markers on tumour-resident cDC1s [30,128]. Interestingly, as seen with CpG-ODN, in a model of B cell lymphoma, G100 induced a protective CD8^+^ T cell response that was dependent on tumour cell expression of TLR4 [129].

It is widely known that exposure to LPS can polarise macrophages to an M1 phenotype [130,131,132]. Others have shown that TLR4 signalling in the TME can promote the recruitment of macrophages from the systemic circulation into the tumour environment [133,134,135]. However, these effects have not been consistently seen with intratumoural treatment. A minor increase in macrophage infiltrate within the TME was observed following G100 administration, but not with LPS [30,128]. Interestingly, low dose LPS resulted in increased infiltration of M1-type macrophages and elevated type I IFN, but failed to control tumour growth in a model of 4T1 breast cancer [136]. Instead, this treatment enhanced tumour growth, resulting in larger primary tumour volume and greater metastasis to the lungs. As intratumourally LPS treated mice displayed elevated systemic inflammatory cytokines, including IL-1α, IL-1β, and TNF, it was suggested that systemic inflammation hindered the antitumour response. Indeed, in a separate study, intratumoural LPS was found to enhance the antitumour response of an oncolytic virus targeting OVA in a model of B16.OVA melanoma, but this combination actually resulted in rapid morbidity and mortality in the majority of mice [137]. Together these findings suggest that strategies to improve sequestration in the tumour with limited leakage are needed, or perhaps the evaluation of TLR4 agonists with less potent activity.

Combination treatments may be required to extract benefit from intratumoural treatment with TLR4 agonists. The combination of intratumoural MPL and agonistic anti-CD40, which is known to activate macrophages, enhanced macrophage-induced killing of tumour cells, resulting in a significant increase in anti-tumour activity in mice compared to either treatment alone [36]. The alarmin, HMGN1, has been identified as a TLR4 ligand, and when injected intratumourally with cyclophosphamide and the TLR7/8 ligand, resiquimod (R848), caused the elimination of established CT26 colon carcinomas [138]. This response correlated with an increase in tumour infiltrating T cells and increased activation and homing of tumour DCs to the draining lymph node. Systemic administration of G100 has been reported to enhance the antitumour response when combined with either anti-PD1 or anti-PDL1 in a model of B16.F10 melanoma [37]. However, the impact of intratumoural delivery has not to our knowledge been reported. Despite this, numerous clinical trials are currently investigating the intratumoural effects of G100 as a monotherapy or in combination with checkpoint inhibitors, radiotherapy, and antigen (NCT02035657, NCT02180698, NCT02320305, NCT02501473).

#### 2.1.5. TLR7/8

The intracellular TLR7/TLR8 receptors recognise single-stranded RNA and RNA viruses, and induce signalling via the MyD88 pathway. The activation of TLR7/8 on cDCs leads to their maturation and improves antigen cross-presentation via increased formation of MHC-peptide complexes, up-regulation of costimulatory molecules (CD80/86, CD40), and release of IL-12, while ligation by TLR7/8 agonists on pDCs induces the production of type I IFNs, further amplifying antigen cross-presentation [139].

The two dual TLR7/8 agonists, resiquimod and telratolimod (also known as MEDI9197 and 3M-052), have been observed to delay tumour growth when administered intratumourally in models of squamous cell carcinoma [140], melanoma [39], colon adenocarcinoma [29], breast cancer [38], and glioma [141]. Both promote polarisation of the TME towards a Th1 phenotype, with increases in effector T cells and M1-type macrophages and the production of IFN-α by pDCs alongside DC maturation. Intratumoural PD-1/PD-L1 expression was increased following telratolimod administration and consequently, the combination of telratolimod and systemic anti-PD-L1 led to increases in median survival in a B16.OVA model and induced tumour regression [29]. Furthermore, combination with anti-PD1 increased survival and prevented metastases in the triple negative 4T1 breast cancer model [38]. Combining telratolimod with both anti-CTLA-4 and anti-PD-L1 resulted in superior activity against treated and untreated tumours than single agents in B16.F10 melanomas, with both tumours displaying an increase in antigen-specific CD8^+^ T cells [39]. Significant tumour growth inhibition was achieved when combining intratumoural telratolimod with systemic agonistic anti-OX40 compared to either agent given alone, which may be due to observed increases in OX40 receptor expression in treated tumours as a result of TLR7/8 agonism [29]. As observed with TLR9 agonists, attempts to improve the retention of TLR7/8 agonists in the tumour by formulating resiquimod into thermosensitive liposomes have been investigated [142].

Telratolimod has been evaluated in a phase I trial in patients with advanced solid tumours either as a monotherapy or in combination with anti-PD-L1 therapy and/or palliative radiation therapy. However, no patient responses were observed, despite both systemic and intratumoural immune activation, suggesting further investigation into the patient populations or combinations that best work with TLR7/8 agonists is needed [143].

### 2.2. STING, RIG-I, or NLR Agonists

#### 2.2.1. RIG-I

RIG-I is a cytosolic PRR that plays a prominent role in antiviral defence by detecting viral and endogenous RNA and triggering binding to the mitochondrial antiviral signalling protein MAVS [144]. Its activation induces apoptosis preferentially in tumour cells and simultaneously activates the innate immune system via type I IFN signalling [145,146]. Intratumoural administration of the RIG-I agonist, MK461, resulted in a rapid antitumour response dependent on NK cells followed by long-term tumour control, most likely mediated by T cells [74,147]. In support, the RIG-I agonist stem loop RNA 14 (SLR14) was found to induce a strong anti-tumour response at both treated and untreated tumour sites, with residual antitumour activity observed in RAG^−/−^ mice [49]. Therefore, both T cells and non-T cells appear to be involved in the response triggered by intratumoural RIG-I signalling. Interestingly, fluorescence-labelled SLR14 is mainly taken up by CD11b^+^ myeloid cells in the TME, however, further investigation into what subtype of myeloid cells and how this enhanced the inflammatory profile in the tumour is required.

An identified advantage of therapy utilising RLR signalling is that tumour cells are highly sensitive to RLR-induced apoptosis, whereas non-malignant cells are protected by endogenous B-cell lymphoma-extra large (Bcl-xL) expression [145,148]. Therefore, RIG-I agonists have been combined with checkpoint inhibitors, with the addition of the agonist overcoming tumour resistance to anti-PD-1 and anti-CTLA4 by enhancing tumour cell death and cross-priming by cDC1s to potentiate systemic cytotoxic T lymphocyte (CTL) antitumour responses [49,149].

#### 2.2.2. STING

STING is a transmembrane receptor that responds to cyclic dinucleotide (CDN) binding, resulting in downstream signalling involving TBK1 activation, IRF-3 phosphorylation, and the production primarily of IFN-β, as well as IFN-α, TNF, and IL-1β [150,151,152,153]. Intratumoural administration of a mouse-selective STING agonist, 5,6-dimethylxanthenone-4acetic acid (DMXAA), was shown to generate potent T cell responses in multiple tumour models, including B16.SIY and methylcholanthrene (MCA)-induced sarcomas, resulting in tumour rejection in the majority of mice [154,155]. A significant portion of the anti-tumour response induced by DMXAA was lost in IFNAR-deficient mice and Batf3^−/−^ mice, supporting the role of cDC1s and type I IFN signalling in STING agonist activity [155]. The response was abrogated in Rag^−/−^ mice, indicating a crucial antitumour T cell response [52]. 

STING activation increases expression of OX40 and CD27 in the TME, while also upregulating inhibitory immune checkpoints such as the PD-1/PD-L1 axis and CTLA-4 [52], suggesting STING agonists should be used in combination with immune checkpoint inhibitors. Indeed, one study found that the agonist, cyclic di-GMP (CDG), potentiated the effects of a triple combination including anti-CTLA-4, anti-PD-1, and agonistic anti-CD137, in a model of prostate cancer [50]. Effective therapy was correlated with increased Teff/Treg ratio and increased M1-type macrophages. A separate study found that STING agonism in a combination with chemotherapeutic carboplatin and anti-PD1 significantly increased survival in an ovarian carcinoma model relative to the carboplatin and anti-PD-1 double combination [51]. 

The synthetic CDNs, 2′3′-c-di-AM(PS)2 (ADU-S100, also called MIW815 or RR-CDA) and 3′3′-cGAMP (cGAMP) and GSK532, and CDN-unrelated STING agonists, TTI-10001 and CRD5500 (also called LB-061), activate human STING alleles and have exhibited antitumour efficacy when administered intratumourally in several mouse tumour models including CT26 colon [53,156,157], MC-38 colon [158], Lewis lung carcinoma [52], and 4T1 breast carcinomas [159]. Similar to studies with murine agonists, induction of a strong type I IFN signal within the TME was central to the response alongside an increase in the Teff/Treg ratio. The dosing regimen was found to important in one study, with repetitive low doses of ADU-S100 shown to be more effective than a high dose regimen at inducing a tumour-specific CD8^+^ T cell response and durable anti-tumour immunity [53]. When used in combination therapy, STING agonists were found to help remodel the tumour vasculature, with the triple combination of ADU-S100 with vascular endothelial growth factor receptor 2 (VEGFR2) blockade and either anti-PD1 or anti-CTLA4 resulting in improved antitumour activity and overall survival [52]. These strong antitumour effects have provided the rationale for clinical assessment of intratumoural STING agonists alone (NCT03937141; NCT04220866; NCT03843359) or in combination with anti-PD1 and anti-CTLA4 (NCT03172936; NCT02675439; NCT03010176).

### 2.3. Available Local Antigen Dose—The Missing Ingredient?

The key threshold in determining the efficacy of intratumoural treatment with PRR agonists is the initial priming (or boosting) of sufficient numbers of T cells to elicit an effective response. Efficacy may, therefore, be down to the antigen dose released within the immunostimulatory environment that is created by treatment. As already noted, antigen dose can be enhanced with the use of agents that induce ICD, and this is likely to be the reason intratumoural PRR agonists often work in synergy with conventional treatments, such as chemotherapy or radiation therapy. This also likely explains the success in preclinical models in combining intratumoural PRR agonists with antibodies that cause antibody-dependent cellular cytotoxicity, such as anti-CD20 and anti-Mucin-1 [160,161,162]. In exploring the concept of raising local antigen dose, an early study showed enhanced antitumour activity with intratumoural CpG-ODN in combination with tumour lysate-pulsed DCs [163]. In another study, CpG-ODNs were combined with a recombinant adenovirus encoding tumour necrosis factor-related apoptosis-inducing ligand (TRAIL) to induce tumour cell apoptosis and thereby release antigens [164]. Absorption of LPS onto a GM-CSF-secreting whole tumour cell vaccine (GVAX) enhanced the therapeutic efficacy when injected intratumourally compared to GVAX alone in a CT26 colon carcinoma model [165]. Where tumour antigens are known, it may be possible to inject specific tumour antigens to increase the local dose. A recombinant adenovirus encoding a tumour-associated antigen was used successfully with CpG-ODN to increase survival times [166], and CpG-ODN also supported induction of antigen-specific responses following injection of a fusion protein containing a viral oncoprotein antigen [167]. Studies incorporating CpG-ODN and antigen into an injectable hydrogel were recently followed up with the additional incorporation of immune cells, which enhanced the antitumour effect [168]. In another study of note, tumours were injected with CpG-ODN and antigen that was not even expressed in the tumour [169]. In this case, the choice of a particularly immunodominant epitope may have generated memory CD8^+^ T cells that performed immunologic helper functions. The authors speculated that other irrelevant antigens may be used similarly, and had previously shown that injecting the pan MHC class II-binding peptide (pan HLA-DR reactive epitope; PADRE) helped control tumours through the generation of CD4^+^ T helper cells [170]. As there have been significant advances in using bioinformatics to quickly develop personalised vaccines against patient-specific neoantigens [171], it is possible that in the future these vaccines could be repurposed for intratumoural delivery with PRR agonists.

With adoptive T cell therapies finding clinical application, current protocols may also be increased by timely intratumoural administration of PRR agonists. In mouse models, the efficacy of transfer of melanoma-specific CTL therapy was increased in this manner, with the PRR agonists activating host cells to enhance the IFN-γ production and killing by adoptively transferred T cells [35]. Intratumoural injection may also be used to help increase tumour-specific T cell numbers in vivo for subsequent ex vivo expansion and transfer [172].

We note that anti-tumour responses to PRR agonists alone have tended to be observed in tumour models known to respond to checkpoint inhibitors, such as CT26, MC38, and 4T1 [29,147,173], suggesting that some level of pre-existing immunity is advantageous. However, as noted earlier, combining checkpoint blockade with PRR agonists can often result in more animals responding, suggesting that the PRR agonists can to some extent turn a “cold” tumour, with limited immune infiltration, into a “hot” one with increased infiltration. While this is could be a major benefit of the combined approach, a potential side effect could be increased toxicity, which is not insignificant with checkpoint blockade alone. However, increased toxicity has not been noted in the clinical application of intratumoural CpG-ODN and anti-PD1 [174], suggesting that the localised effect of intratumoural administration may limit deleterious effects. Nonetheless, the dose of PRR agonists used in the clinic should be considered carefully, especially where repeated dosing in the context of combined treatment could potentially amplify risks.

## 3. Concluding Statement

Preclinical studies have highlighted the unique added benefit of delivering PRR agonists directly into the tumour bed. However, full regression is rarely seen with this treatment alone, which has also been seen in clinical studies. The future of this form of treatment is therefore likely to be in combination with other cancer therapies, whether this is chemotherapy, radiation therapy, or recently developed immunotherapies based on T cell modulation. The major advantage of such “in situ vaccination” with PRR agonists is that the tumour antigens targeted do not have to be defined. Nonetheless, it may be possible to incorporate this untargeted element of the treatment with targeted therapies, be they antigen-specific or personalised vaccines, or adoptive T cell therapies.

## Figures and Tables

**Figure 1 cancers-12-03824-f001:**
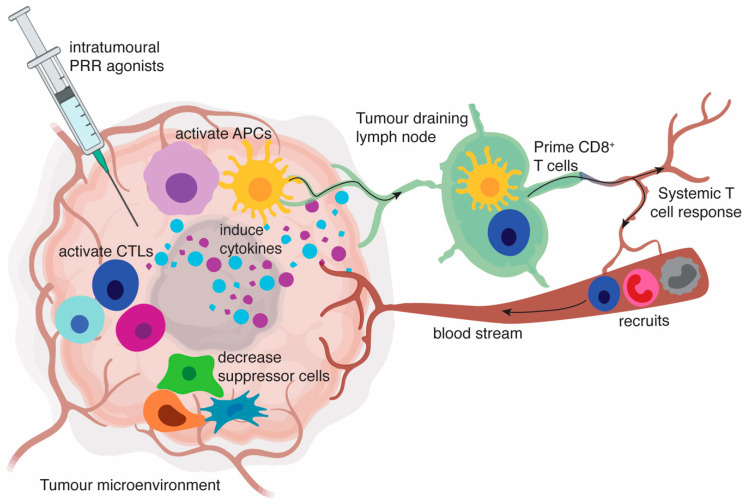
Intratumoural pattern recognition receptors (PRR) agonists induce a proinflammatory tumour microenvironment. PRR engagement on antigen-presenting cells (APCs) results in the activation of APCs including conventional myeloid dendritic cells (cDCs), plasmacytoid dendritic cells (pDCs) and macrophages. These activated APCs produce cytokine profiles mostly dominated by type I interferons (IFN) as well as proinflammatory cytokines, such as IL-12p70, IL-6, IL-1 and tumour necrosis factor (TNF). This creates an environment that favours the activation or restimulation of T cells, decreases the activity of myeloid-derived suppressor cells (MDSCs) and regulatory T cells (Tregs), and polarises macrophages to an anti-tumoural M1-type phenotype. Activated APCs also appear in tumour-draining lymph nodes, where they can contribute to presentation of tumour-associated antigens to T cells. Chemokines released in the tumour help recruit circulating T cells, natural killer (NK) cells, monocytes, and neutrophils. Ultimately, a systemic immune response is generated that can be capable of inducing regression at distant tumours.

**Table 1 cancers-12-03824-t001:** Intratumoural PRR agonists and effective treatment combinations in a preclinical setting.

Intratumoural Agonist	Analogues	PRR Receptors	Notable Cytokines	Effects on APCs	Critical Effector Cells	Beneficial Combinations
Pam_3_CSK_4_		TLR1 TLR2	IFN-γ	Limited effects alone	Limited effects alone	Anti-CTLA-4 [33]
acGM-1.8		TLR2	TNF, IL-12p70 and IFN-γ	Increased M1/M2 macrophage ratio	Increased Teff/Treg ratio	
Poly I:C	BO-112	TLR3 RIG-I	Type I IFN	Dependent on cDC1	Increased Teff/Treg ratio	Flt3 ligand with anti-PD-L1 mAb and BRAF inhibition [34]; CpG-ODN [35]
Poly A:U		TLR3	Type I IFN	Activate cDCs	Increased Teff/Treg ratio and granzyme B-producing CD8^+^ T cells	
LPS	MPL G100	TLR4	Type I IFN	Activate cDC1s Increased M1/M2 macrophage ratio	Increased Teff Increased NK cells	Anti-CD40 mAb [36]; anti-PD1 or anti-PD-L1 [37]
Resiquimod (R848)		TLR7 TLR8	Type I IFN	Activate pDCs Activation of cDCs Increased M1/M2 macrophage ratio	Increased Teff	
Telratolimod		TLR7 TLR8	Type I IFN	Activate pDCs Activation of cDCs Increased M1/M2 macrophage ratio	Increased Teff	Anti-PD-L1 mAb [29]; anti-PD-1 mAb [38]; anti-CTLA-4 and anti-PD-L1 mAb [39]; anti-OX40 mAb [29]
CpG ODN	SD-101 CMP-001 IMO-2125	TLR9	Type I IFN	Macrophages responsible for early rejection phase Activate pDCs Accumulation and activation of cDCs Reduced MDSC function	Increased infiltration of antigen-specific T cells Increased NK cells	Anti-CD40 antibody [40]; anti-CTLA4 [41]; anti-OX40 mAb alone [42] or in combination with anti-CTLA-4 [43]; anti-PD-1 mAb alone [44] or in combination with local radiotherapy [45]; metronomic cyclophosphamide [31], doxorubicin [46]; ibrutinib [47]; poly I:C [35]; 3M-052 [48]
MK461		RIG-I	Type I IFN		Dependent on T and NK cells	
SLR14		RIG-I	Type I IFN	Phagocytosed by CD11b^+^ myeloid cells	Increased Teff/Treg ratio	Anti-PD-1 [49]
Murine CDNs	DMXAA CDG	STING	Type I IFN	Dependent on cDC1	Dependent on T cells	Anti-CTLA-4, anti-PD-1 and agonistic anti-CD137 [50]; carboplatin and anti-PD1 [51]
Synthetic human CDNs	ADU-S100 cGAMP GSK532	STING	Type I IFN	Activate cDCs	Dependent on CD8^+^ T cells	VEGFR2 blockade and anti-CTLA-4 or anti-PD1 mAb [52]; anti-PD-1 with anti-CTLA4 [53]

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
