# Peer review of "Modulating the Tumour Microenvironment by Intratumoural Injection of Pattern Recognition Receptor Agonists"

_cancers, 2020, doi:10.3390/cancers12123824_

Round 1
Reviewer 1 Report
In the review, the authors summarized the pattern recognition receptors (PRRs) including TLR2-9, RIG-I, STING and their agonists as well as their roles in tumor microenvironment after intratumoral administration. They discussed the effects of these PRR agonists in regulating different immune cells, inflammatory factors and immune checkpoints in tumor microenvironment. This is a very interesting review paper.
Comments:
- The authors summarized the agonists for TLR2-9 and STING/RIG-I/NLR, respectively. If the authors can provide a detailed Table including the important information that described in the text for all the agonists of TLR2-9 and STING/RIG-I/NLR, it will be very helpful for reader to clearly understand each agonist. The specificity for agonist is very important. Some agonists may target for multiple receptors, such as agonists for TLR7/8, and are the rest of agonists all special for one receptor or several? The authors can include the specificity in the Table.
Author Response
We would first like to thank the reviewer for their time and comments on our manuscript.
In response to the reviewer's suggestion, we have added a table to the paper showing the receptors targeted by each of the agonists described in the text, and the key effects each agonist induces as a single agent, plus notable combination treatments.
Reviewer 2 Report
Olivia K Burn, et al. summarized the effects of pattern recognition receptors (PRRs) with intratumoral injection on enhancement of the activity of antigen presenting cells and adaptive immune responses against tumor. Intratumoral administration of PRRs is a promising method to fight cancers and has potential to contribute to new strategies for cancer treatment. The article is well written, and the statement are comprehensible. There are a couple of questions.
・In this article, a large variety of intratumoral PRR ligands is explained. I would like to know optimal selection of PRR ligands for variation of targeted tumors. Are therapeutic effects after intratumoral administration of PRRs associated with immune cell component in tumor microenvironment of targeted tumors?
・Direct injection of PRR agonists into the tumors is required due to toxicities associated with systemic administration. Can intratumoral administration of PRRs be performed repeatedly? How should the repetition of the intratumoral injection be managed?
・The authors state that full regression is rarely seen with this treatment alone and the future of this form of treatment is likely to be in combination with other cancer therapies. What kind of adverse effects can newly occur by combination with other cancer therapies?
Author Response
We would first like to thank the reviewer for their time and comments on our manuscript.
Q1: “I would like to know optimal selection of PRR ligands for variation of targeted tumours. Are therapeutic effects after intratumoural administration of PRR associated with immune cell component in tumour microenvironment of targeted tumours?”
A1: If we have interpreted this question correctly, the reviewer is asking how can PRR agonists be selected for use in specific cancer indications. We have not specifically addressed this issue in the review. The work described is preclinical only, relying heavily on syngeneic models, and although these models are somewhat representative of specific indications, we do not believe they should be over-interpreted. Perhaps an exception is lymphoma, where we have already noted in the text that there is a unique response to CpG-ODN, with the lymphoma cells functioning as APCs themselves through TLR stimulation; this same mechanism may be expected to occur in the clinic, perhaps providing an advantage to CpG-ODN over other agonists in this clinical setting.
If we have interpreted the second part of the question correctly, the reviewer is asking whether the intratumoural treatments rely on pre-existing immune infiltrates before treatment. In many of the sections we describe how the effects are mediated through local immune cells, either through reprogramming their own tumouricidal activities, or stimulating capability to recruit other effector cells. We have now added a small summary section noting that, in general, efficacy has been seen in preclinical models where there is some evidence that checkpoint blockade can work (even if only to a limited degree), which does suggest that pre-existing immunity is advantageous (lines 518-520).
Q2: Can intratumoural administration of PRRs be performed repeatedly? How should the repetition of the intratumoural injection be managed?
A2: We have now noted in the text (line 82) that many studies have used a repetitive dosing regimen without increased toxicity. However, we have also cautioned that “repeated dosing in the context of combined treatment could potentially amplify risk” (lines 527-529 ), so needs to be considered carefully. We have also noted the importance of assessing the dose of the agonist if it is to be repeatedly administered, giving an example where repetitive low doses of a STING agonist was shown to be more effective than a high dose regimen at inducing a tumour-specific CD8+ T cell response and durable anti-tumour immunity (lines 455-457).
Q3: What kind of adverse effects can newly occur by combination with other cancer therapies?
A3: To address this question, we have added the following text (lines 518-529):
“We note that anti-tumour responses to PRR agonists alone have tended to be observed in tumour models known to respond to checkpoint inhibitors, such as CT26, MC38 and 4T1, suggesting that some level of pre-existing immunity is advantageous. However, as noted earlier, combining checkpoint blockade with PRR agonists can often result in more animal responding, suggesting that the PRR agonists can to some extent turn a “cold” tumour, with limited immune infiltration, into a “hot” one with increased infiltration. While this is could be a major benefit of the combined approach, a potential side effect could be increased toxicity, which is not insignificant with checkpoint blockade alone. However, increased toxicity has not been noted in the clinical application of intratumoural CpG-ODN and anti-PD1, suggesting that the localised effect of intratumoural administration may limit deleterious effects. Nonetheless, the dose of PRR agonists used in the clinic should be considered carefully, especially where repeated dosing in the context of combined treatment could potentially amplify risk.”